# A BioID-Derived Proximity Interactome for SARS-CoV-2 Proteins

**DOI:** 10.3390/v14030611

**Published:** 2022-03-15

**Authors:** Danielle G. May, Laura Martin-Sancho, Valesca Anschau, Sophie Liu, Rachel J. Chrisopulos, Kelsey L. Scott, Charles T. Halfmann, Ramon Díaz Peña, Dexter Pratt, Alexandre R. Campos, Kyle J. Roux

**Affiliations:** 1Enabling Technologies Group, Sanford Research, Sioux Falls, SD 57104, USA; danielle.may@sanfordhealth.org (D.G.M.); rachel.chrisopulos@sanfordhealth.org (R.J.C.); kelsey.scott@sanfordhealth.org (K.L.S.); charles.halfmann@sanfordhealth.org (C.T.H.); 2Proteomics Facility, Sanford Burnham Prebys Medical Discovery Institute, La Jolla, CA 92037, USA; lmartin@scripps.edu (L.M.-S.); vanschau@sbpdiscovery.org (V.A.); rdiaz@sbpdiscovery.org (R.D.P.); 3Division of Genetics, Department of Medicine, University of California, San Diego, CA 92093, USA; sol015@health.ucsd.edu (S.L.); depratt@ucsd.edu (D.P.); 4Department of Pediatrics, Sanford School of Medicine, University of South Dakota, Sioux Falls, SD 57069, USA

**Keywords:** SARS-CoV-2, proximity labeling, BioID, TurboID, COVID-19, interactome

## Abstract

The novel coronavirus SARS-CoV-2 is responsible for the ongoing COVID-19 pandemic and has caused a major health and economic burden worldwide. Understanding how SARS-CoV-2 viral proteins behave in host cells can reveal underlying mechanisms of pathogenesis and assist in development of antiviral therapies. Here, the cellular impact of expressing SARS-CoV-2 viral proteins was studied by global proteomic analysis, and proximity biotinylation (BioID) was used to map the SARS-CoV-2 virus–host interactome in human lung cancer-derived cells. Functional enrichment analyses revealed previously reported and unreported cellular pathways that are associated with SARS-CoV-2 proteins. We have established a website to host the proteomic data to allow for public access and continued analysis of host–viral protein associations and whole-cell proteomes of cells expressing the viral–BioID fusion proteins. Furthermore, we identified 66 high-confidence interactions by comparing this study with previous reports, providing a strong foundation for future follow-up studies. Finally, we cross-referenced candidate interactors with the CLUE drug library to identify potential therapeutics for drug-repurposing efforts. Collectively, these studies provide a valuable resource to uncover novel SARS-CoV-2 biology and inform development of antivirals.

## 1. Introduction

The 2019 novel coronavirus, SARS-CoV-2, is the causative agent of Coronavirus Disease 2019 (COVID-19) and responsible for a global pandemic. COVID-19 most often presents as a respiratory illness, yet can cause gastrointestinal and/or neurological symptoms and acute cardiac injury as well [1,2,3]. Presently, hundreds of millions of people have been infected with SARS-CoV-2 worldwide, and several million people have died as a result. Long-term effects of COVID-19 infection are reported by 10–30% of patients, and as millions of people recover from COVID-19, questions remain about vertical transmission of COVID-19 infection during pregnancy and post-COVID syndrome symptoms, including pulmonary fibrosis, neurological defects, and vascular dysfunction [4,5,6,7,8,9,10]. While wide-spread vaccination is likely to slow the spread of COVID-19, developing treatment strategies for new infections and long-term post-COVID symptoms will require a thorough understanding of the SARS-CoV-2 virus and how it affects patient cell biology.

A crucial component of the effort to study COVID-19 is the application of technologies that reveal how viral proteins behave in host cells. Current efforts to map the SARS-CoV-2 virus-host interactome have offered great insight into possible pathways directly affected by various viral proteins, yet differences in experimental approaches and data analysis methods inevitably lead to discrepancies when comparing reported interactomes [11,12,13,14,15,16,17,18,19,20,21,22]. As with any large-scale approach to identifying gene or protein networks, false positives due to background contamination can hinder accurate data interpretation; therefore, the use of several approaches with multiple replicates by multiple independent studies will be required to ultimately map the full SARS-CoV-2 interactome.

Proximity-dependent labeling of host proteins via BioID or similar promiscuous biotin ligases fused to viral proteins has been used to study host–viral protein associations for a number of viruses, including herpes simplex virus type 1, Epstein–Barr virus, Zika virus, Ebola virus, and coronaviruses [23,24,25,26,27]. Here, we generated A549 human lung cancer cells stably expressing BioID-tagged SARS-CoV-2 viral proteins to identify whole-cell proteomic changes due to viral protein expression and to identify specific protein–protein interactions (PPIs) between individual SARS-CoV-2 viral proteins and host-cell proteins. We compared our BioID datasets with similar available proximity-based proteome datasets to develop a list of high-confidence candidate protein interactors. Finally, we cross-referenced our BioID dataset with the CLUE drug library of clinical and FDA-approved drugs to identify potentially beneficial drugs for COVID-related treatments. Collectively, these datasets comprise an invaluable resource for COVID biologists, as we present several focused avenues of future exploration and provide a Shiny application for further independent analysis by investigators. All of our data are available at (https://alexproteomics.shinyapps.io/covid19proteomics/, accessed on 4 March 2022) for further in-depth analysis by the scientific community, and will serve to further our collective understanding of infection mechanisms, contribute to current drug repurposing efforts, and guide higher-confidence follow-up studies investigating specific PPIs and pathway alterations.

## 2. Materials and Methods

### 2.1. Plasmids

SARS-CoV-2 viral genes from the Wuhan-Hu-1 (MN975262) and 2019-nCoV/USA-WA1/2020 (MN985325) isolates were amplified via PCR from Addgene (Watertown, MA, USA) constructs with a 1x (GGGS) linker incorporated into each primer set (see Appendix A). Amplified PCR products were fused to biotin ligases via In-Fusion Recombination (Takara Bio Inc., Kusatsu, Shiga, Japan) into myc-BioID2 pBabe (Addgene #80900; XhoI/PmeI), BioID2-HA pBabe (Addgene #120308; BamHI/EcoRI), or TurboID-3xHA pBabe (BamHI/EcoRI) [28]. mycBioID2 (Addgene #80900) was used as a control for BioID2 cell lines. Human albumin signal sequence-3xHA-TurboID-KDEL pBabe control construct was made by two-step In-Fusion Recombination. Human albumin signal sequence and 3xHA-TurboID [28] were PCR-amplified with KDEL built into the reverse primer. Fragments were inserted into myc-BioID2 pBabe, replacing mycBioID2 (Addgene #80900; EcoRI/PmeI). Placing of N- or C-terminus tags was based on a previous report with two exceptions [11]. NSP3 (not studied in [11]) was tagged on the N-terminus due to multiple transmembrane regions and protease activity. NSP12 was tagged on the N-terminus, because when expressed with a C-terminus tag the overall expression and biotinylation activity was insufficient. All fusion-protein plasmids will be made available on Addgene.

### 2.2. Cell Culture

A549 cells were obtained from the American Type Culture Collection (ATCC, Manassas, VA, USA; CCL-185™). Stable cell lines for all constructs were generated using retroviral transduction. HEK293 Phoenix cells (National Gene Vector Biorepository, Indianapolis, IN, USA) were transfected with each construct using Lipofectamine 3000 (Thermo Fisher Scientific, Waltham, MA, USA) per manufacturer′s recommendation. The transfected cells were incubated at 37 °C for 6 h. After 6 h incubation, the transfected cells were replenished with fresh medium and further incubated at 32 °C for 72 h. The culture media were filtered through a 0.45-μm filter and added to A549 cells along with Polybrene (4 μg/mL; Santa Cruz Biotechnology, Dallas, TX, USA). At 72 h after transduction, puromycin (0.5 μg/mL; Thermo Fisher Scientific) was added to the target cells for 72 h and viable cells were pooled. The expression of fusion proteins and functional biotinylation following addition of 50 µM biotin was further verified using IF and WB. The stable cell lines were maintained in 5.0% CO_2_ at 37 °C in DMEM (HyClone, Logan, UT, USA) supplemented with 10% fetal bovine serum (FBS). All cells were tested monthly for mycoplasma contamination.

### 2.3. Immunofluorescence

Cells grown on glass coverslips were fixed in 3% (wt/vol) paraformaldehyde/phosphate-buffered saline (PBS) for 10 min and permeabilized by 0.4% (wt/vol) Triton X-100/PBS for 15 min. For labeling fusion proteins, chicken anti-BioID2 (1:5000; BID2-CP-100; BioFront Technologies, Tallahassee, FL, USA) or mouse anti-hemagglutinin primary antibody was used (HA; 1:1000; 12CA5; Covance, Princeton, NJ, USA). The primary antibody was detected using Alexa Fluor 568–conjugated goat anti-chicken (1:1000; ab175477, Lot#GR144853-2, Abcam, Cambridge, United Kingdom) or Alexa Fluor 568–conjugated goat anti-mouse (1:1000; A11004; Lot#1698376, Thermo Fisher Scientific). Alexa Fluor 488–conjugated streptavidin (1:1000; S32354; Lot#2201616, Thermo Fisher Scientific) was used to detect biotinylated proteins. DNA was detected with Hoechst dye 33342. Coverslips were mounted using 10% (wt/vol) Mowiol 4-88 (Polysciences, Warrington, PA, USA). Epifluorescence images were captured using a Nikon Eclipse NiE (40 ×/0.75 Plan Apo Nikon objective, Minato City, Tokyo, Japan) microscope.

### 2.4. Western Blot Analysis

To analyze total cell lysates by immunoblot, 1.2 × 10^6^ cells were lysed in SDS-PAGE sample buffer, boiled for 5 min, and sonicated to shear DNA. Proteins were separated on 4–20% gradient gels (Mini-PROTEAN TGX; Bio-Rad, Hercules, CA, USA) and transferred to nitrocellulose membrane (Bio-Rad). After blocking with 10% (vol/vol) adult bovine serum and 0.2% Triton X-100 in PBS for 30 min, the membrane was incubated with appropriate primary antibodies: chicken anti-BioID2 (1:5000; BID2-CP-100; BioFront Technologies, Tallahassee, FL, USA) or rabbit polyclonal anti-hemagglutinin (1:2000; Ab9110; Lot#GR218331-6, Abcam). The primary antibodies were detected using horseradish peroxidase (HRP)–conjugated anti-chicken (1:40,000; A9046; Lot#015M4856V, Sigma-Aldrich, St. Louis, MO, USA) or anti-rabbit (1:40,000; G21234; Lot#2156243, Thermo Fisher Scientific). The signals from antibodies were detected using enhanced chemiluminescence via a Bio-Rad ChemiDoc MP System (Bio-Rad, Hercules, CA, USA). Following detection of each antibody, the membrane was quenched with 30% H^2^O^2^ for 30 min. To detect biotinylated proteins, the membrane was incubated with HRP-conjugated streptavidin (1:40,000; ab7403; Lot#GR305788-2, Abcam) in 0.2% Triton X-100 in PBS for 45 min.

### 2.5. Sample Preparation

BioID pulldown for each cell line was performed in triplicate, with distinct samples for each replicate. For each large-scale BioID2 pulldown sample, two 10 cm dishes at 80% confluency were incubated with 50 µm biotin for 18 h, washed twice with PBS, and cell pellets collected for automated BioID pulldown at the Proteomics Facility at Sanford Burnham Prebys Medical Institute (La Jolla, CA, USA). TurboID samples were prepared similarly, except they were treated with 50 µm biotin for only 4 h. Briefly, cells were lysed in 8 M urea, 50 mM ammonium bicarbonate (ABC) and benzonase and the lysate was centrifuged at 14,000× *g* for 15 min to remove cellular debris. Supernatant protein concentration was determined using a bicinchoninic acid (BCA) protein assay (Thermo Scientific). Disulfide bridges were reduced with 5 mM tris(2-carboxyethyl)phosphine (TCEP) at 30 °C for 60 min, and cysteines were subsequently alkylated with 15 mM iodoacetamide (IAA) in the dark at room temperature for 30 min. Each sample was separated into two aliquots, one for whole cell proteome profiling and the other for proximity-dependent labeling analysis. Whole cell protein lysate was digested overnight with mass spec-grade Trypsin/Lys-C mix (1:25 enzyme/substrate ratio). Following digestion, samples were acidified with formic acid (FA) and subsequently desalted using Agilent (Santa Clara, CA, USA) AssayMap C18 cartridges mounted on an Agilent AssayMap BRAVO liquid handling system. Cartridges were sequentially conditioned with 100% acetonitrile (ACN) and 0.1% FA, samples were then loaded, washed with 0.1% FA, and peptides eluted with 60% ACN, 0.1% FA. Finally, the organic solvent was removed in a SpeedVac (Thermo Fisher Scientific) concentrator prior to LC-MS/MS analysis.

### 2.6. Affinity Purification of Biotinylated Proteins

Affinity purification and digestion of biotinylated proteins were carried out in an automated fashion in a Bravo AssayMap platform (Agilent) using AssayMap streptavidin cartridges (Agilent). Briefly, cartridges were first primed with 50 mM ammonium bicarbonate and then proteins were slowly loaded onto the streptavidin cartridge. Background contamination was removed with 8 M urea, 50 mM ammonium bicarbonate. Finally, cartridges were washed with Rapid digestion buffer (Promega, Madison, WI, USA, Rapid digestion buffer kit) and proteins were subjected to on-cartridge digestion with mass spec grade Trypsin/Lys-C Rapid digestion enzyme (Promega) at 70 °C for 1 h. Digested peptides were then desalted in the Bravo platform using AssayMap C18 cartridges and dried down in a SpeedVac concentrator.

### 2.7. Mass Spectrometry

Prior to LC-MS/MS analysis, dried peptides were reconstituted with 2% ACN, 0.1% FA and concentration was determined using a NanoDropTM spectrophometer (Thermo Fisher Scientific). Samples were then analyzed by LC-MS/MS using a Proxeon EASY-nanoLC system (Thermo Fisher Scientific) coupled to an Orbitrap Fusion Lumos mass spectrometer (Thermo Fisher Scientific). Peptides were separated using an analytical C18 Aurora column (75 µm × 250 mm, 1.6 µm particles; IonOpticks, Fitzroy, Victoria, Australia) at a flow rate of 300 nL/min (60 °C) using a 75-min gradient: 1% to 5% B in 1 min, 6% to 23% B in 44 min, 23% to 34% B in 28 min, and 34% to 48% B in 2 min (A = FA 0.1%; B = 80% ACN: 0.1% FA). The mass spectrometer was operated in positive data-dependent acquisition mode. MS1 spectra were measured in the Orbitrap in a mass-to-charge (m/z) of 375–1500 with a resolution of 60,000 at m/z 200. Automatic gain control target was set to 4 × 10^5^ with a maximum injection time of 50 ms. The instrument was set to run in top speed mode with 2 s cycles for the survey and the MS/MS scans. After a survey scan, the most abundant precursors (with charge state between +2 and +7) were isolated in the quadrupole with an isolation window of 0.7 m/z and fragmented with HCD at 30% normalized collision energy. Fragmented precursors were detected in the ion trap as rapid scan mode with automatic gain control target set to 1 × 10^4^ and a maximum injection time set at 35 ms. The dynamic exclusion was set to 20 s with a 10 ppm mass tolerance around the precursor. The mass spectrometry raw data and search results files generated in this study are available in the ProteomeXchange (proteomexchange.org, accessed on 4 March 2022) and were uploaded via MASSIVE with the dataset identifier PXD029207 and MSV000088245, respectively.

### 2.8. Data Analysis

All raw files were processed with MaxQuant (version 1.5.5.1) using the integrated Andromeda Search engine against a target/decoy version of the curated human Uniprot proteome without isoforms (downloaded in 2 January 2020) and the GPM cRAP sequences (commonly known protein contaminants). First search peptide tolerance was set to 20 ppm, main search peptide tolerance was set to 4.5 ppm, and fragment mass tolerance was set to 20 ppm. Trypsin was set as enzyme in specific mode and up to two missed cleavages were allowed. Carbamidomethylation of cysteine was specified as fixed modification and protein N-terminal acetylation and oxidation of methionine were considered variable modifications. The target decoy-based false discovery rate (FDR) filter for spectrum and protein identification was set to 1%.

Statistical analysis of interactome data was carried out using in-house R script (version 3.5.1, 64-bit), including R Bioconductor (Boston, MA, USA) packages such as limma and MSstats. First, peptide intensities were log2-transformed and loess-normalized using the normalizeCyclicLoess function from the limma package (version 3.46.0) with all default parameters across replicates of each bait or control batch to account for systematic errors. Note that normalization was not carried out across all samples due to significant differences in pulldowns of different baits and/or their controls. Testing for differential abundance was performed using MSstats bioconductor package (version 3.22.1) using all default parameters of the groupComparison and dataProcess functions, except the normalization which was set to FALSE, as data were previously normalized as described above. Importantly, the Log2FC and *p*-value of proteins missing in all replicates of the negative controls and detected in at least one of the bait samples (i.e., NSP or ORF) was imputed. The imputation was performed post-statistical test, again only for the proteins that failed to be tested by MSstats because they were completely missing one condition. The imputed Log2FC was calculated as the average of the protein intensity (i.e., sum of peptide intensities of a given protein within a given sample) across the triplicate of the same bait, divided by 3.3. On the other hand, the imputed *p*-value was computed by dividing 0.05 by the number of replicates of a given bait in which the protein was confidently identified. Therefore, the imputed log_2_FC provides a notion of the average protein intensity in a pulldown, while the imputed *p*-value reports the confidence of identification in the sense of reproducibility of detection. These values avoid large numbers and allow for cleaner visualization via volcano plots and other charts/graphs. For example, a potential prey candidate detected in all three replicates of a given bait with log2 intensities of 18.7, 19.5, and 20.3 will have an imputed Log_2_FC of 5.9 and a *p*-value of 0.0167. In contrast, another potential prey candidate detected in two replicates (15.5 and 16.5) will have an imputed Log_2_FC of 4.8 and a *p*-value of 0.025. In addition, we generated a CrapomeScore for each identified protein in the experiment. The CrapomeScore is the fraction of all streptavidin-based experiments in the Crapome database (reprint-apms.org, accessed on 2 January 2020) that the prey protein is identified. The CrapomeScore ranges from 0 to 1, and a protein with a score of 1 means that it was identified in all streptavidin-based experiments in the Crapome database.

### 2.9. Global Proteome Data Analysis

All global proteome datasets were compared to BioID2-only or TurboID-KDEL (used only to compare ORF8-TurboID) control cell lines. Filter cut-offs were set at log_2_FC ≥ 2 (upregulated) or log_2_FC ≤ −2 (downregulated), *p* value ≤ 0.01, at least two quantitative peptide features, and detected in less than 75% of the proximity-labeling CRAPome contaminant database experiments. These parameters were chosen in an attempt to minimize false positives while maximizing true positives.

### 2.10. Network Analysis of SARS-CoV-2 Interactors

A hierarchical model of cellular processes and structures predicted to interact with SARS-CoV-2 was derived via multi-scale community detection performed on a large protein interaction network. We selected a network derived from the STRING database as our starting network: the subset of the STRING interactions with a combined confidence score greater than 0.7 (available in the Network Data Exchange (NDEx) at https://www.ndexbio.org/viewer/networks/275bd84e-3d18-11e8-a935-0ac135e8bacf, accessed on 25 February 2022) [29,30]. Then, human proteins interacting with SARS-CoV-2 proteins were filtered by log_2_FC ≥ 2.32, *p* value ≤ 0.01, and *n* ≥ 2 in an attempt to minimize false positives while maximizing true positives. The specific high-confidence interactions were filtered based on the CRAPome contaminant database with a score ≤0.5 [31]. A subnetwork “proximal” to those proteins then was identified by network propagation using the Cytoscape Diffusion tool [32].

Multi-scale community detection analysis was performed on this subnetwork using the community detection algorithm HiDeF via the Community Detection APplication and Service (CDAPS; app available at http://apps.cytoscape.org/apps/cycommunitydetection, accessed on 21 November 2021) [33,34]. The resulting hierarchical model describes “communities” in the network at multiple scales, where communities are subnetworks of proteins interacting more with each other than with other proteins in the network. The analysis infers a structure to the network, one in which communities are hypotheses for processes or structures that interact with SARS-CoV-2 proteins. The communities are organized into a hierarchy in which larger communities subsume smaller communities [35,36]. Finally, the hierarchy network (https://doi.org/10.18119/N9531R, accessed on 4 March 2022) was styled, communities were subjected to enrichment analysis in GO biological processes using the g:Profiler package in CDAPS, *p* values were calculated based on the hypergeometric distribution, and a layout was applied.

### 2.11. Virus-Centric Analysis of SARS-CoV-2 Interactors

To provide a visual model that displays high-confidence cellular factors that interact with individual SARS-CoV-2 proteins, we utilized the same network derived from the STRING database (confidence score > 0.7), with protein groups that had degree of connection = 1, log_2_FC ≥ 2.32, and *p* value ≤ 0.01. In addition, we filtered for promiscuity using the CRAPome repository (CRAPome ≤ 0.5) and included only proteins that were found in two or more biological replicates [31]. The resulting high-confidence interactors were visualized using Cytoscape (v3.8.0) and tested for enrichment in GO biological process terms using the hypergeometric distribution [37].

### 2.12. Integrated Analysis of Global Proteome and PPI Data for NSP7/8/12

Interaction candidates were filtered as described above, and global abundance changes filtered by log_2_FC ≥ 2 (upregulated) or log_2_FC ≤ −2 (downregulated), *p* value ≤ 0.01, at least two quantitative peptide features, detected in less than 75% of the proximity-labeling CRAPome contaminant database experiments, and significant to only one viral bait (NSP7, NSP8, or NSP12). A hierarchical model of cellular processes and structures was derived via multi-scale community detection performed on the protein interaction network derived from the STRING database. Multi-scale community detection analysis was performed on this network using the community detection algorithm HiDeF via the Community Detection APplication and Service (CDAPS; app available at http://apps.cytoscape.org/apps/cycommunitydetection, accessed on 21 November 2021) [33,34]. The resulting hierarchical model describes “communities” in the network at multiple scales, where communities are subnetworks of proteins interacting more with each other than with other proteins in the network. The analysis infers a structure to the network, one in which communities are hypotheses for processes or structures that interact with viral bait proteins. The communities are organized into a hierarchy in which larger communities subsume smaller communities [35,36]. Finally, the hierarchy network was styled, communities were subjected to enrichment analysis in GO biological processes using the g:Profiler package in CDAPS, and a layout was applied. GO biological processes enrichment on interaction candidates was performed using the Functional Enrichment tool on the COVID19 Proteomics Resource (https://alexproteomics.shinyapps.io/covid19proteomics, accessed on 21 November 2021) utilizing the filter parameters described above.

### 2.13. Integrated Analysis of SARS-CoV-2 Interactome Datasets

Datasets were obtained from three separate studies [14,15,22]. Interactions deemed significant by respective authors were compiled in one excel spreadsheet without filtering out preys identified across multiple baits. “High confidence interactions” were those that were identified in at least three studies and four datasets with degree of connection being ≤3 for at least four datasets.

### 2.14. Web-Based Shiny App

An accompanying web-based Shiny application (https://alexproteomics.shinyapps.io/covid19proteomics, accessed on 4 March 2022) was created to allow visualization and further functional analysis of the BioID and whole cell proteome statistical analysis data. The application uses several applications, including clusterProfiler (v3.18.1), for functional analysis with the *enricher* function, using the Broad Institute molecular signature databases (v7.4) including canonical pathways (Reactome, KEGG, WikiPathways http://www.gsea-msigdb.org/gsea/msigdb/, accessed on 4 March 2022), immune collection, chemical and genetic perturbation signatures, regulatory transcription factor targets (TFT), oncogenic signatures, and Gene Ontology (Human Phenotype, Cellular Component, Biological Process and Molecular Function). Using the CLUE’s Drug Repurposing Hub database from the Broad Institute (version 3/24/2020), we annotated all the compounds known, according to our data, to target host proteins that potentially bind to viral proteins.

## 3. Results

### 3.1. Development of Stable BioID Cell Lines Expressing Individual SARS-CoV-2 Viral Proteins

The SARS-CoV-2 virus generates two long polypeptides that are cleaved into sixteen non-structural proteins (NSPs) as well as several downstream ORFs encoding four structural proteins (Spike, Envelope, Membrane, and Nucleocapsid, or S, E, M, and N) and nine accessory proteins. In order to identify global cellular changes associated with viral protein expression as well as to identify specific viral-host PPIs, the promiscuous biotin ligase BioID2 was fused to either the N- or C- terminus of individual SARS-CoV-2 proteins and stably expressed by retroviral transduction in human lung cancer A549 cells (Appendix A, Figure 1A). For each construct, we included a GGGS linker to alleviate steric hindrance between the BioID ligase and viral protein. Each cell line was validated by immunofluorescence (IF) and western blot (WB) for fusion-protein expression and biotinylation, revealing a wide-range of permissible expression levels and overall biotinylation (Figure 1B, Appendix A). Three proteins (Spike, Nsp1, and ORF3d) were excluded from this study due to an inability to generate cells stably expressing BioID2-fusion proteins, leaving 26 viral-BioID2 fusion proteins. ORF8, a predicted lumenal protein, was tagged with the TurboID ligase that was previously shown to be substantially more active in the ER lumen compared to BioID [28]. BioID2-alone was used as a control for these viral protein fusions, with the exception of ORF8, for which we utilized a signal sequence-TurboID-KDEL (TurboID-KDEL) to target and retained the ligase in the ER-lumen. Each cell line was processed in triplicate and subjected to whole-cell lysis for global proteome analysis and affinity purification of biotinylated proteins for identification of PPIs via mass spectrometry.

### 3.2. SARS-CoV-2 Proteomics Website

All datasets were collected to build a COVID-19 Proteomics Resource for the scientific community. We created an interactive ShinyApp website to disseminate and explore the functional landscapes of SARS-CoV-2 viral protein interactomes and proteomes. At https://alexproteomics.shinyapps.io/covid19proteomics, accessed on 4 March 2022, we have made all global abundance and proximity-labeling MS data publicly available along with several tools to enable statistical and bioinformatics analysis. The website allows users to interactively explore the data, easily set confidence thresholds, and run functional enrichment analysis using, for example, a hypergeometric test against the Broad Institute molecular signature databases (v7.4) including canonical pathways (Reactome, KEGG, WikiPathways), immune collection, chemical and genetic perturbation signatures, regulatory transcription factor targets (TFT), oncogenic signatures, and Gene Ontology (Human Phenotype, Cellular Component, Biological Process and Molecular Function). In addition, users can compare functional enrichment of defined groups of viral proteins using the compareCluster function of the clusterProfiler R Bioconductor package. Our website allows for the comparison of our BioID dataset with three other interactome datasets, and uses the CLUE drug library to annotate potential therapeutic targets specifically identified in our BioID data. Finally, all data are available to download in spreadsheet form if the user wishes to further supplement their analysis with other available tools (e.g., STRING, Metascape). Examples of website functionality are shown in Figure 2.

### 3.3. Whole-Proteome Analysis of Cells Overexpressing Individual BioID-Viral Bait Fusion Proteins

Unlike previous SARS-CoV-2 proximity-labeling studies, we both identified PPIs for each viral protein and performed global proteome analysis to identify changes associated with expression of individual SARS-CoV-2 viral proteins (Appendix A). To exclude false positives due to ligase expression, we compared global proteome changes in cells expressing individual viral fusion-proteins to their respective control cell lines (those expressing ligase only) and filtered proteins by log_2_FC ≥ 2 (upregulated) or log_2_FC ≤ −2 (downregulated), *p*-value ≤ 0.01, at least two quantitative peptide features, and detected in less than 75% of the proximity-labeling CRAPome contaminant database experiments. Not surprisingly, we saw a marked increase in proteins involved in cytokine signaling in the immune system (HSA-1280215) in response to viral protein expression, including CD70, IRF9, and TNFSF9. The most significantly upregulated protein we identified was ITGB3 (logFC = +3.32 to +4.78), which has recently been shown to be upregulated in COVID-19 patient lung samples and has been hypothesized to be an alternative receptor for the SARS-CoV-2 virus [38,39]. We found this ITGB3 upregulation in cells expressing ORF9c, ORF3a, ORF7b, E-protein, and NSP2. Interestingly, the most significantly downregulated proteins (logFC = −2.60 to −9.94) were MUC5AC and MUC5B, with a dramatic reduction in cells expressing viral NSP12, NSP15, ORF7b, NSP3, NSP2, E, ORF9c, and ORF3a. Levels of these proteins were significantly reduced in cells expressing NSP5, NSP6, NSP14, and N-protein as well, although to a lesser extent. MUC5AC/B are proteins involved in mucus secretion in the respiratory tract, and these data suggest that several of the SARS-CoV-2 proteins are capable of globally reducing cellular MUC5AC/B proteins, even when expressed individually. Proteins involved in DNA replication processes were significantly suppressed, especially by NSP2 expression, including BRCA1 (NSP2, logFC = −2.58), PRIM2 (NSP2, logFC = −2.56), and CDCA2 (NSP2, logFC = −2.40), suggesting that NSP2 may play a key role in directly and/or indirectly disrupting cell cycle progression and apoptosis pathways.

The Membrane (M-), Nucleocapsid (N-), and Envelope (E-) proteins are three of the four structural proteins of SARS-CoV-2, and are known to interact with each other [40]. Both M- and E-protein are membrane proteins important for viral entry, and N-protein is the primary RNA-binding protein involved in properly packaging viral RNA into new vesicles [41]. In order to take a closer look at how SARS-CoV-2 structural proteins influence host-cell protein expression, global changes were visualized via volcano plot (Figure 3A) and the 24 upregulated and 57 downregulated proteins from Appendix A were analyzed via Metascape express analysis (Figure 3B,C). The top two significantly enriched terms identified for upregulated proteins were regulation of actin cytoskeleton (hsa04810; *p* > 10^−4^) and RHO GTPase effectors (R-HAS-195258; *p* > 10^−3^) (Figure 3B), suggesting mechanisms by which SARS-CoV-2 remodels cytoskeletal networks for viral entry and budding. Additionally, three subsets of protein–protein interaction networks were identified, including several proteins involved in differentiation [42,43,44,45,46]. We identified significantly enriched terms in the group of downregulated proteins, including cellular hormone metabolic process (GO:0034754; *p* > 10^−5^) and maintenance of gastrointestinal epithelium (Figure 3C). Further investigation of these proteins could elucidate how viral infection hijacks cellular metabolism and/or causes symptoms of gastrointestinal distress. Taken together, these whole-cell proteome datasets will act as an important foundation to direct potential future investigational studies.

### 3.4. Network Analysis of SARS-CoV-2 Host Interactors Reveals Novel Biology

In addition to whole-cell proteomic analysis, each stable cell line was subjected to BioID proximity-labeling to identify specific viral-host PPIs. Following statistical test, we identified 3011 significant viral-host PPIs, with a log_2_FC ≥ 2.3, *p* value ≤ 0.01, at least two quantitative peptide features, and detected in less than 75% of the proximity-labeling CRAPome contaminant database experiments. This list of significant PPIs is an available resource on our COVID-19 Proteomics Resource website, allowing users to interactively explore networks and functions of the detected PPIs. To understand the functional and biochemical relationships between the identified SARS-CoV-2 interactors, we conducted hierarchy and pathway enrichment analyses (see Methods) on a subset of 876 proteins uniquely associated with one of the 26 SARS-CoV-2 proteins (Appendix A) with CrapomeScore ≤0.5 (i.e., detected in 50% or less of the CRAPome proximity-labeling experiments; see Methods). These analyses revealed that most identified SARS-CoV-2 interactors were associated with seven clusters that included host translation machinery, endocytosis and vesicle transport, metabolism, glycosylation, cell junctions and ion transport, maintenance of homeostasis, and mitochondrial function (Figure 4A). Subclusters within host translation included processing of mRNAs and non-sense mediated decay (NMD) (*p* = 1.13^−36^), which is involved in degradation of aberrant self and non-self mRNAs, including those of coronaviruses [47]. Consistent with previous systems-level studies of SARS-CoV-2, a significant number of the interactors were associated with endocytosis and vesicle trafficking pathways, including members of the SNARE complex, which are important for membrane fusion of vesicles and exocytosis [48], as well as GTPases that regulate vesicle docking and likely support SARS-CoV-2 trafficking and egress. Notably, we found a highly enriched cluster of SARS-CoV-2 interactors involved in cholesterol biosynthesis (*p* = 1.13^−24^) (Figure 4B), providing further evidence of the importance of this pathway for SARS-CoV-2 replication and highlighting potential targets for therapeutic efforts [49]. SARS-CoV-2 interactors were associated with mitochondrial function, including proteins of the TIM/TOM complex that mediate mitochondrial import (*p* = 1.44^−11^) and proteins involved in the electron transport chain (*p* = 2.93^−14^) and oxidative phosphorylation (*p* = 3.32^−7^), which could reflect the energetic requirements of SARS-CoV-2 for replication [50]. Notably, our analysis revealed several pathways involved in cell junctions and ion transport. These included members of the SWELL complex (Figure 4C), which are involved in transport of cGAMP generated upon activation of the immune sensor cGAS by DNA viruses or mtDNA release [51], as well as proteins involved in cell adherens junctions (*p* = 2.01^−5^), previously shown to be targeted by viruses to alter the environment of bystander cells and suggested as therapeutic targets to prevent viral spread [52]. In addition, amongst novel SARS-CoV-2 interactors were several members of the ABC-transporter family (*p* = 7.58^−12^) (Figure 4D), involved in translocation of substrates across membranes and previously linked to development of multidrug resistance (MDR) and oxidative stress response to viral and bacterial infection [53]. Proteins interacting with SARS-CoV-2 were associated with the peroxisome (*p* = 1.88^−6^). SARS-CoV-2 infection has been shown to recruit peroxisomes to viral replication organelles, and the association of SARS-CoV-2 with members of the peroxisome could reflect a requirement to reduce oxidative stress resulting from the extensive remodeling of cellular endomembranes or as a lipid source for viral replication [54,55].

### 3.5. Focused Analysis of Individual Viral–Host Protein Interactions

To identify the relationships between discrete SARS-CoV-2 proteins and cellular functions, we conducted pathway analyses on the cellular PPI candidates for each viral protein (see Methods). Here, we report on the relationships identified for four SARS-CoV-2 viral proteins; however, all BioID data and several tools for pathway analysis have been made available on the COVID-19 Proteomics Resource website, as described above.

#### 3.5.1. ORF3a

We identified 68 unique interactors for protein ORF3a, a viroporin involved in viral replication and release, 37 of which were transmembrane proteins, including endosomal, lysosomal, and other vesicular proteins [56]. SARS-CoV-2 utilizes deacidified lysosomes for egress and, consistent with this process, ORF3a interactors revealed enrichment in lysosomal transport proteins, regulators of endosome and lysosome fusion, and regulators of pH and ion homeostasis (Figure 5A) [57]. As previously reported, we identified HOPS endosomal tethering complex proteins VPS11 and VPS39, as well as WWP1, a HECT ubiquitin ligase that has been previously associated with viral budding via the VPS pathway [11,58]; we also identified previously unreported ORF3a interactors to be involved in cell adhesion and adherens junctions, which could be exploited by SARS-CoV-2 to control cell-to-cell communication and promote cell spread. These results suggest that ORF3a plays a multifaceted role during viral infection including a major role in membrane reorganization and trafficking, perhaps specifically utilizing the HECT/VPS viral budding pathway to enhance viral release.

#### 3.5.2. ORF6

We identified 50 candidate interactors for ORF6, a membrane-associated protein reported to localize to the ER [59]. In line with previous studies, we identified SEC24A/B, proteins associated with COPII-coated vesicle transport, and other SEC complex proteins involved in ER homeostasis as associated with ORF6 (Figure 5B) [14,15]. Other ORF6-associated proteins included cell cycle regulators of G1 to S phase transition GSPT1 and GSPT2, as well as PYCR1, PYCR2, and RRM2B, previously linked to cell cycle arrest at G1 phase [60]. Several RNA viruses manipulate critical cell cycle regulators or induce cell cycle arrest to favor viral replication, including inhibition of early apoptosis in infected cells, evasion of immune defenses, or to promote assembly of viral particles [61]. Additionally, several novel ORF6 interactors involved in deubiquitylation were identified, potentially suggesting a mechanism for deubiquitylation of viral proteins to evade degradation at the proteosome or by autophagy. Alternatively, ORF6 could influence deubiquitylation pathways to stabilize cellular factors that are supportive of viral replication, including USP5, which acts as a negative regulator of type I IFN signaling and has been found to increase in abundance during SARS-CoV-2 replication [62].

#### 3.5.3. ORF8

The ORF8 protein has been implicated in modulating innate and adaptive immune response, specifically via downregulation of MHC-I [63,64]. Furthermore, deletion of SARS-CoV-2 ORF8 leads to a decrease in proinflammatory cytokine release and increases efficacy of immune response in COVID-19 patients [65]. In line with this, we identified 64 unique PPIs, including six proteins involved in type I IFN signaling and six proteins involved in O-linked glycosylation (Figure 5C). These data support previous work linking the O-linked glycosylation process with ORF8, which could serve to evade the immune system using molecular mimicry and glycan shielding [11,16,66,67]. Previously unreported ORF8 associations include several factors of the innate immune response, including OAS1, OASL, MX1, and PLSCR1, all of which are implicated in negative regulation of viral genome replication (GO:0045071), potentially supporting ORF8 as a key regulator of host immune response during SARS-CoV-2 infection. Additionally, we observed novel ORF8 associations with proteins implicated in MAPK signaling (KRAS, LGALS3, LGALS8, and ARRB2) and dephosphorylation process (MTMR1, MTMR2, and MTMR10); *p* = 4.31^−4^. These proteins play a role in intracellular membrane trafficking, and vesicle transport [68], and may thus serve to establish a mechanism for viral spread by controlling cell signaling, replication, and survival.

#### 3.5.4. NSP4

Coronavirus NSP4 is part of the viral replication complex and rearranges host cell membranes to induce double-membrane vesicles for viral replication [69]. Our BioID analysis identified 112 protein candidates uniquely associated with NSP4, including proteins involved in membrane lipid biosynthesis pathways, glycerphospholipid metabolism, and members of the N-glycan precursor biosynthesis machinery (Figure 5D). NSP4 was associated with proteins involved in ubiquitination and proteosome degradation (CUL1, HERC2, and ANAPC2; *p* = 4.27^−4^), as well as previously unreported associations with members of the ER-associated protein degradation (ERAD) pathway (SEC61B, SEC62, ANAPCP2 and MARCH6; *p* = 8.10^−8^), suggesting a potential mechanism by which viral proteins can evade host-degradation machinery. Association of NSP4 with ERAD proteins could suggest antagonism of ERAD-mediated degradation of viral proteins by, for instance, autophagy, or an attempt to manipulate ERAD pathway to degrade immune regulators with antiviral properties in order to facilitate viral trafficking and release [70].

### 3.6. Integrated Analysis of PPI Networks and Global Abundance Changes among the Viral Proteins That Make up the RdRp Replication Complex

The SARS-CoV-2 RNA-dependent RNA polymerase (RdRp) complex required for viral genome transcription and replication is composed of the non-structural proteins NSP7, NSP8, and NSP12. To perform an integrated analysis of the overall changes associated specifically with the subunits of the RdRp complex, we performed community detection analysis on the proteins uniquely enriched in BioID pulldowns, as well as uniquely up- or downregulated proteins detected in each cell line (Figure 6A). We identified several communities of interest, including polyadenylation-dependent snoRNA 3′-end processing. This group contained EXOSC2/4 (interaction candidates, NSP12), HBS1L (interaction candidate, NSP8) and DDX60 (upregulated, NSP12), proteins involved in RNA exosome response to RNA-virus infection [71,72]. Additionally, we found IGFBP3 was specifically downregulated in cells expressing NSP12, a protein with levels that correlate with adult respiratory distress syndrome severity [73,74]. GO-BP enrichment of NSP7/8/12 PPI candidates identified protein groups involved in RNA surveillance and processing, mitochondrial transport, and myofibril assembly (Figure 6B). Collectively, these data present several interesting avenues of exploration for COVID biologists specifically studying the SARS-CoV-2 RdRp complex.

### 3.7. Integrated Analysis with Previously Published Datasets

#### 3.7.1. SARS-CoV-2 Interaction with the Cellular Restrictome

To further explore the interplay of the identified SARS-CoV-2 interactors and the innate immune response, we leveraged a recent gain-of-function screen that identified 65 interferon stimulated genes (ISG) that act to inhibit SARS-CoV-2 replication [75]. Cross-comparison between these two datasets revealed that seven of these ISGs were found in association with one or more SARS-CoV-2 proteins, including ISG15, IFIT1, IFIT5, IFITM2, IFITM3, MLKL, and SPATS2L (Appendix A). Our study revealed an association between viral N and SPAT2SL, an ISG that was found to inhibit SARS-CoV-2 RNA replication and is involved in formation of stress granules [75,76]. SARS-CoV-2 has been suggested to antagonize stress granules to evade immune responses, and these data suggest that N and SPAT2SL interaction could be important for this mechanism [77,78]. SARS-CoV-2 ORF9c has been recently associated with evasion of immune responses, though the molecular regulators are yet to be defined [79]. Our study elucidated Or9c in association with IFIT5, an ISG that targets non-self RNA for degradation and was found to inhibit SARS-CoV-2 replication [75], thus suggesting that this factor could be targeted by ORF9c for immune evasion. Finally, ORF8b and ORF9b have been shown to trigger mechanisms of cell death [80]. Consistent with those findings, ORF9c was found to associate with the activator of necroptosis MLKL [81]. More work will be required to characterize these factors and investigate their role in SARS-CoV-2 pathogenesis.

#### 3.7.2. Utilizing Previous SARS-CoV-2 BioID Interactome Datasets to Develop a List of High-Confidence Interactions

In order to develop a high-confidence list of viral-host PPIs, we compared our BioID dataset to three previously published proximity-labeling datasets [14,15,22]. “High confidence interactions” were those that were identified in at least three studies and four datasets, with degree of connection ≤ 3 for at least four datasets. We were able to curate a list of 66 viral-host PPIs across seventeen SARS-CoV-2 viral proteins identified by at least three of these reports, with seventeen total interactions identified by all four studies (Appendix A). Seventeen ORF9b PPIs were identified by at least three reports, including the antiviral signaling protein MAVS and mitochondrial fusion/fission proteins MFF, MTFR1L, and USP30. We further substantiated ORF6 involvement in ER to Golgi vesicle transport via manipulation of COPII complex (SEC31A), with a possible role in mediating COPI-mediated transport via interaction with the coatomer protein ARCN1. Recently a fourth epidemic wave of COVID-19 in Hong Kong was attributed to a mutation in ORF3a, and another ORF3a mutation was previously associated with higher mortality rate [82,83]. We identified four high-confidence ORF3a interactions, including late endosome membrane protein VPS39, as well as endosomal adaptor protein NUMB/L, which could serve as potential therapeutic targets. Altogether, this comparison of similarly produced datasets sets a firm foundation for higher-confidence follow-up studies based on interaction evidence put forth by four separate interactome studies.

#### 3.7.3. Cross-Referencing PPI Interactions with the CLUE Drug Library for Drug Repurposing Efforts

Finally, we used the CLUE drug library to cross-reference potential therapeutic targets revealed by our BioID drug screens with clinical and FDA-approved drugs to aid future drug-repurposing efforts [84]. Based on the unique candidates identified for all viral baits found in all three triplicates per bait, we identified 48 total host-protein targets with 211 known FDA-approved drug interactions with the potential for COVID-19 therapeutic use (Appendix A). Our dataset supports previous COVID-19 drug repurposing studies and can increase the confidence in certain drugs consistently proposed for repurposing, including the JAK1 inhibitor Baricitinib [85], retinoic acid receptor agonist Acitretin [86], and ATPase inhibitors digoxin and ouabain [87]. However, significant follow-up investigation will be necessary to confirm the impact these potential drugs may have on full SARS-CoV-2 infection. A drug repurposing visualization tool has been made available on our COVID-19 Proteomics website for specific and multiplexed analysis of viral baits, host-proteins, and corresponding drug interactors.

## 4. Discussion

Previous attempts to map the SARS-CoV-2 viral interactome have varied in experimental approach, data analysis parameters, cell lines used, and specific viral baits [11,12,13,14,15,16,18,19,20,21]. Unlike previous BioID reports, we report here both global proteome analysis and BioID-based proximity interactome analysis in human A549 lung cells for all but three SARS-CoV-2 viral proteins. Utilizing these complementary datasets, we analyzed several individual viral proteins and viral protein groups using various methods as examples for future avenues of exploration and compare these data to previously reported COVID-BioID datasets to identify consistently reported candidates. The specific parameters and thresholds used in this report are presented as examples, and certainly do not exhaust all cogent avenues of statistical analysis. Therefore, we present these data as a resource, and developed a website to host the data to allow for more in-depth analysis of global proteomic changes in response to individual viral proteins, analyze enriched interaction candidates, pathway enrichment bioinformatics analysis, and comparative analysis to other reported datasets including the ISG SARS-CoV-2 inhibitors, previous COVID-BioID datasets, and the CLUE drug library for drug repurposing. While this report primarily discusses unique identifications to highlight the highest-confidence interactors, the COVID-19 Proteomics Resource website (https://alexproteomics.shinyapps.io/covid19proteomics, accessed on 4 March 2022) will allow for multiplexed and variable in-depth analysis of the MS data presented here. This study complements previous proximity interactome studies and drug repurposing identification efforts by strengthening confidence in reported interactors and proposed drug treatments, as well as identifying new interactors and previously unidentified drug candidates.

In this study, we profiled 26 of the 29 known SARS-CoV-2 viral proteins. Consistent with previous studies, we noted substantial suppression of NSP1 translation in our NSP1-BioID2 stable cell line (data not shown) and therefore chose to exclude NSP1 from this study [15,88]. Surprisingly, we were unable to successfully clone the Spike protein, which is one of the most widely-studied SARS-CoV-2 proteins, and chose not to pursue troubleshooting Spike-BioID so as not to delay dissemination of this data and due to the extensive characterization of Spike already underway (see [89] for review). Finally, the hypothetical ORF3d protein (previously referred to as ORF3b [11,90]) was excluded due to our inability to generate cell lines stably expressing this protein. The use of human A549 lung-cancer derived cells for these studies is both a strength and a limitation. These cells do retain certain fundamental traits of alveolar type-II pulmonary epithelial cells; however, A549 cells are not clearly representative of normal human pulmonary epithelial cells. An additional limitation of these studies is that while the BioID analysis of viral proteins in isolation allows for identification of specific PPIs, it precludes potential viral protein–complex interactions that would occur with full SARS-CoV-2 infection.

Our global profiles of human lung cells overexpressing individual SARS-CoV-2 viral proteins produced a large dataset of significantly upregulated or downregulated cellular proteins, enabling the ability to identify specific viral proteins influencing specific changes in cell biology. While the expression of individual viral proteins likely does not impact the cell similar to SARS-CoV-2 infection, this approach does allow for focused analysis and exploration of individual viral protein function. These data support previous reports of ITGB3 overexpression in SARS-CoV-2 infected cells and tissues, and further identifies the specific viral proteins that could be influencing the overexpression. If ITGB3 is indeed working as an alternate receptor for SARS-CoV-2 viral uptake, it may be that targeting ITGB3 or the specific viral proteins that upregulate ITGB3 levels could have therapeutic benefit to slow cell-to-cell spread of the virus. Additionally, our findings that several of the SARS-CoV-2 proteins can reduce cellular levels of MUC5AC/B, possibly via increased secretion [91,92], provides insight into one of the mechanisms by which the virus causes devastation of the respiratory system in the most severe COVID-19 cases. Most importantly, this global proteome data can be analyzed with each respective proximity labeling dataset to investigate how protein–protein interactions are affecting specific pathways within the cell on a larger scale. While we briefly analyzed global proteome data on two sets of interacting viral proteins (structural and RdRp viral proteins), further in-depth analysis by dedicated virologists will be necessary to validate these findings, especially in the context of active SARS-CoV-2 infection. This resource will act as a platform for COVID biologists to perform integrated analyses and identify interesting and significant objectives for further investigation.

While previous interactome studies have reported PPI candidates even when identified in up to six viral protein interactomes [14,15], we highlighted here only unique protein candidates for each viral bait in order to reduce the possibility of capturing promiscuous interactors and keeping in mind that due to its limited coding capacity, RNA viruses have likely very little functional redundancy within their genomes [93,94]. For this reason, and for brevity, we chose to pursue a strict analysis for this report, predominantly focusing on PPIs uniquely identified for each bait. Unfortunately, the strict thresholds used for this report returned no significant interactors for NSP5 or NSP10; however, further analysis of the data utilizing the website could yield true interactors. While this approach should allow for the identification of high-confidence interactors, it is important to note that many of the viral proteins reside in the same subcellular compartments and would thus likely be proximate to many of the same proteins, leading to their detection. However, it may be that proteins identified by more than one viral bait do have unique biological relevance to at least one viral bait; thus, we have enabled iBAQ intensity analysis on the website to allow for more in-depth comparative analysis of the MS data to reveal these more substantial associations.

Our data further support previously published studies including the role of ORF3a in extensive membrane remodeling and viral budding via interaction with VPS39 and VPS11, and suggest potential novel interactions between Orf3a and cell adhesion factors, which are important for cell-to-cell communication. In line with previous studies, our ORF6 data support interaction with SEC-complex proteins and suggests novel roles in cell cycle regulation and viral immune evasion via deubiquitination mechanisms [14,15]. ORF8 is known to play a role in immune evasion, and our data support a possible role in viral immune evasion via O-linked glycosylation and suggest immune signaling disruption via interaction with effectors and regulators of the type I IFN response. Our data demonstrate clear support of previously published reports, and our novel findings implicating new roles for SARS-CoV-2 viral proteins allow for an even more comprehensive understanding of how SARS-CoV-2 interacts with host cells.

As variants arise and COVID-19 infections continue to threaten lives and cause lingering effects through post-COVID syndrome, the need for a clear SARS-CoV-2 viral–host interactome has never been more evident. The ability to identify crucial viral–host interactions and potentially disrupt those interactions with drug repurposing would allow for fast-tracked treatments to be made available to those suffering from COVID-19 and long-term symptoms. The data within this resource alone do not confer biological relevance to each proteomic change or protein interaction identified, and as such more studies are needed to verify the viral–host interactions presented here. Similarly, considerable studies will be necessary to assess whether and how any of the pharmacological agents identified by the CLUE library analysis affect biological processes involving SARS-CoV-2. However, the specific bioinformatics performed here provide several confident avenues of future exploration, and the COVID-19 Proteomics Resource website will serve to help guide researchers and provide high-confidence directions for future studies.

## Figures and Tables

**Figure 1 viruses-14-00611-f001:**
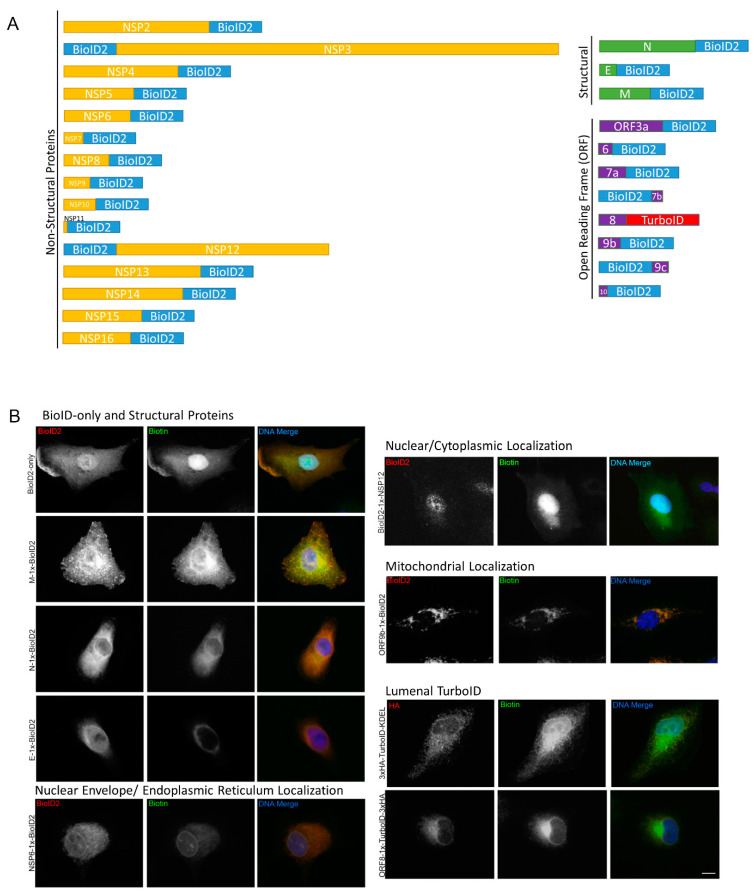
BioID2-viral fusion protein expression in A549 cells. (**A**) Viral proteins were fused to either the N- or C-terminus of the BioID2 promiscuous biotin ligase. The schematic shows the orientation of NSPs (yellow), structural proteins (green), and ORF proteins (purple) fused to BioID2 to scale. (**B**) A549 human lung cells stably expressing BioID2-fusion proteins were assessed for fusion-protein expression and localization (red) and promiscuous biotinylation (green) following the addition of exogenous biotin. Scale bar 10 µm.

**Figure 2 viruses-14-00611-f002:**
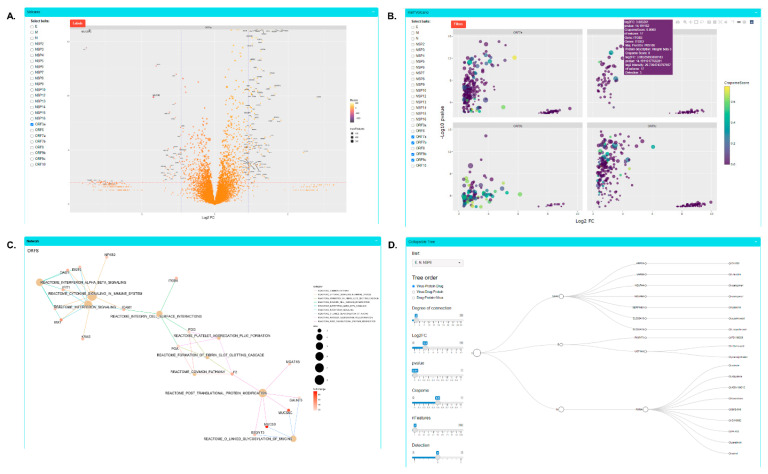
Examples of COVID-19 Proteomics website functionality. (**A**) Volcano plot analysis of changes in global protein abundance. (**B**) Half volcano plots showing enriched PPI candidates following BioID method. (**C**) Functional enrichment analysis of PPI candidates. (**D**) Protein–drug tree for viral proteins, PPI candidates, and known drug interactors.

**Figure 3 viruses-14-00611-f003:**
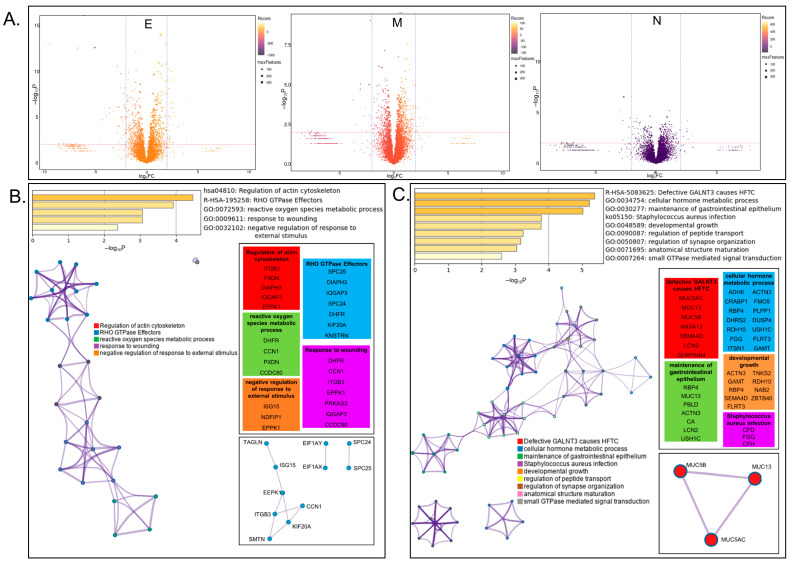
Analysis of proteins influenced by expression of viral structural E, M, and N proteins. (**A**) Volcano plot visualization of global changes in host-protein expression in cells expressing viral proteins Envelope (E), Membrane (M), or Nucleocapsid (N). Protein identifications are available on the COVID-19 Proteomics Resource website. (**B**) Metascape analysis of significantly upregulated proteins including enriched terms, cluster visualization, and interactions. (**C**) Metascape analysis of significantly downregulated proteins including enriched terms, cluster visualization, and interactions.

**Figure 4 viruses-14-00611-f004:**
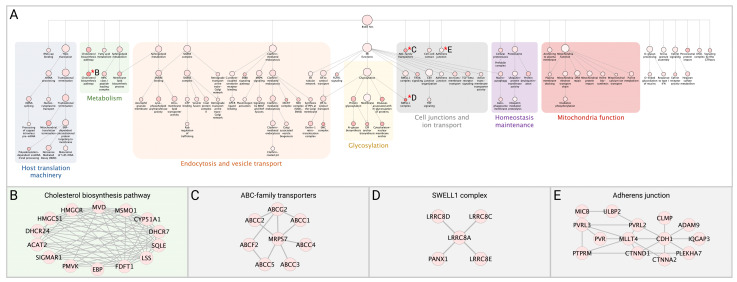
Network analysis of SARS-CoV-2 interactors. (**A**) The network containing the 876 identified SARS-CoV-2 interactors was subjected to supervised community detection, and the resulting hierarchy is shown. Each node represents a cluster of interconnected proteins and each edge (marked by an arrow) represents containment of one community (target) by another (source). Indicated are enriched biological processes as determined by gProfiler. (**B**–**E**) Asterisks (*) denote selected zoom-in insets from the hierarchy. Nodes represent human proteins, and edges are interactions from STRING.

**Figure 5 viruses-14-00611-f005:**
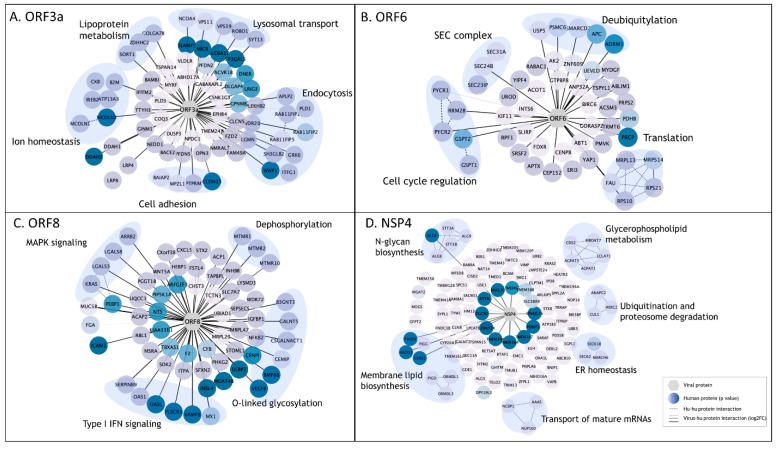
Enriched pathway analysis of PPIs for selected SARS-CoV-2 viral baits. (**A**–**D**). High-confidence associations between indicated SARS-CoV-2 proteins (hexagons) and human proteins (circles/nodes). Node color is proportional to the *p* value (the darkest, the lowest the *p* value). Human–human interactions as determined by STRING are represented by dashed edges. Human–viral interactions are indicated with solid edges, and their thickness are proportional to the log_2_FC (the thickest, the highest log_2_FC).

**Figure 6 viruses-14-00611-f006:**
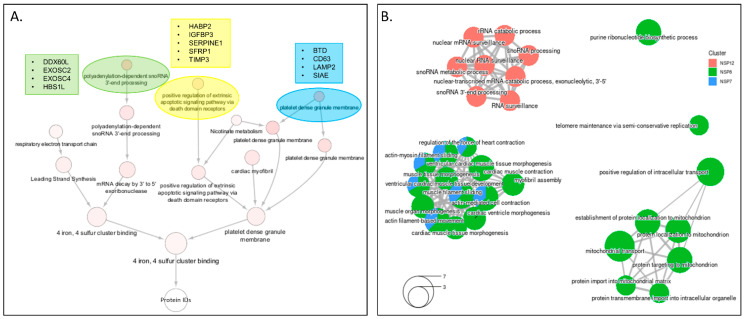
Integrated bioinformatics analysis of datasets from cells expressing individual RDRP-complex viral proteins. (**A**) The network containing the 123 proteins identified as NSP7/8/12 unique interactors, globally upregulated, or globally downregulated was subjected to supervised community detection and the resulting hierarchy is shown. Each node represents a cluster of interconnected proteins and each edge (marked by an arrow) represents containment of one community (target) by another (source). Indicated are enriched biological processes as determined by gProfiler. (**B**) Gene ontology enrichment of biological processes performed by unique host-proteins identified in proximity labeling studies for NSP7/8/12.

## Data Availability

The mass spectrometry raw data and search results files generated in this study are available in the ProteomeXchange and were uploaded via MASSIVE with the dataset identifier PXD029207 and MSV000088245, respectively. The network for SARS-CoV-2 interactions reported here is available in the Network Data Exchange (NDEx) at https://www.ndexbio.org/viewer/networks/275bd84e-3d18-11e8-a935-0ac135e8bacf (accessed on 4 March 2022). Additional datasets and tools are available as a resource on the COVID19 Proteomics Resource (https://alexproteomics.shinyapps.io/covid19proteomics, accessed on 4 March 2022).

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
