# Peer review of "A BioID-Derived Proximity Interactome for SARS-CoV-2 Proteins"

_viruses, 2022, doi:10.3390/v14030611_

Round 1

Reviewer 1 Report

In this manuscript May et al., describe an almost comprehensive proximity interactome analysis of SARS-CoV-2 proteins using the BioID technique. In addition, the outcome of expressing these proteins in the host cells has also been investigated.

The study is well designed and the resulting data are in line with previously published or preprint interactome studies on SARS-CoV-2 proteins but also provide some new insights. A strong point of the present report is the analysis of all but three SARS-CoV-2 proteins, the global proteome analyses upon overexpression of the viral proteins and the compilation of all the generated data in a publicly available shiny app that allows convenient data mining. All in all this study will be a useful resource for the SARS-CoV-2 community.

In my opinion this manuscript is ready for publication as such with a few minor comments to address:

  1. May the authors comment on which basis some viral proteins were fused at the N- and others at the C-terminus with the biotin ligase (functional data / structural prediction / other criteria?)

  1. The statistical analysis pipeline is too superficially described (lines 194-200): could the author display their in-house scripts so that others can use them too, in particular which parameters were used for the loess normalization and what was exactly done with MSstats

  1. The description of the imputation procedure (lines 201-206) in its present form is not easy to understand. Could the authors give an example of an imputation following their procedure. In addition, the rational for choosing the values 3.3 (Log2(10)) and 0.05 in the calculations is not clear to me.

  1. Line 587, COPI is usually written with a roman number (like similar to COPII)

  1. Some of the proteins studies by the authors normally interact together (e.g. M, N and E), may the authors comment if expressing these proteins individually may distort the obtained interactome (e.g. because of missing combined interfaces or a potential different conformation of a protein within a complex)?

Reviewer 2 Report

This manuscript by May et al describes a systematic approach to defining the virus-host protein-protein interactomes of individual SARS-CoV-2 proteins, as well as the effect of individual viral protein expression on the global cellular proteome. The authors generated stable cell lines that expressed genes encoding SARS-CoV-2 proteins fused to BioID tags, and these cell lines were used for global proteomic profiling as well as pulldown studies. Notably, the authors have compiled their data set and made it publicly accessible via an easy to use website which facilitates quantitative target identification and biological contextualization of the data. The report overall is well written, and the methods are exceptional which is important when establishing a data repository such as this. This report should be of interest to the readership of Viruses and the underlying data set/website should be of high value to researchers studying SARS-CoV-2 virology and molecular mechanisms of pathogenesis.

I have provided detailed comments that I hope will further improve the overall manuscript. These include comments aimed at clarifying methodology, data analysis, and visualization. In addition, although the authors are aware that this data set is largely a tool for exploration and hypothesis generations, I have also included several comments intended to guide a more nuanced description of the data, as the text at times does not clearly distinguish between true viral infection and these studies using stable cell lines. Finally, I would caution the authors on their linkage between drug repurposing data and this proteomics study. The data presented in this report does not provide direct evidence of any host protein being a viable therapeutic target that would warrant such investigation. I strongly encourage the authors to state this clearly in the text.
